# Evaluation of Outcomes following Reduction in Targeted Fluid Administration in Major Burns

Maryum Merchant [1],*, Scott B. Hu [1],*, Stella Cohen [1], Peter H. Grossman [2], Kurt M. Richards [2] and Malcolm I. Smith [1]

[1] Department of Pulmonary/Critical Care, University of California Los Angeles, Los Angeles, CA 90095-1405, USA; stellacohen@mednet.ucla.edu (S.C.); ismith@mednet.ucla.edu (M.I.S.)
[2] Grossman Burn Center, West Hills, CA 91307, USA; phg@grossmanmed.com (P.H.G.); kurt@grossmanmed.com (K.M.R.)
* Correspondence: mmerchant@mednet.ucla.edu (M.M.); scotthu@mednet.ucla.edu (S.B.H.)

**Abstract:** Adequate fluid resuscitation in adults with major burns is crucial to prevent or minimize burn shock, but needs to be balanced against the complications of over-resuscitation. A single-center, retrospective review of 95 ICU patients with severe burns from Jan 2014 to Aug 2021 was performed. Some 52 patients were managed with a liberal targeted fluid goal of 4 mL/kg/%TBSA, and 43 patients were managed after we incorporated a restricted fluid goal of 2 mL/kg/%TBSA into our standard resuscitation strategy. Of the 95 patients included in this analysis, 76 patients (80%) survived admission. The median age was 41 years, and the median TBSA was 36%. All patients received Ringer's lactate as the primary fluid for resuscitation, and 40 of the 95 patients (42%) received colloids as a rescue infusion within 24 h of injury. Some 44 of the 95 patients (46.3%) had a concurrent inhalational injury. The median length of hospital stay was 37 days, and the median ICU length of stay was 18 days. A total of 17 of the 95 patients developed ARDS (17.9%), 51 of the 95 (53.7%) patients developed pneumonia, and 34 of the 95 patients (35.8%) developed AKI within the first 7 days of admission. The median fluid administered during the first day of hospitalization from 2019 onwards remained close to 4 mL/kg/%TBSA, despite transitioning to a 2 mL/kg/%TBSA formula for a 24 h fluid goal (unless there was an electrical burn, in which case the 4 cc/kg formula was utilized). Further exploratory analyses also suggested that under-resuscitation and administration of albumin may be associated with increased mortality, though this did not reach statistical significance. ARDS development was associated with increased age and TBSA as well as increased fluid intake within the first 24 h. A change in the targeted fluid goal from liberal (4 mL/kg/%TBSA) to a restricted (2 mL/kg/%TBSA) formula did not change the actual fluids administered over 24 h when guided by clinical criteria. Our review did suggest that under-resuscitation contributed to mortality, but that excessive fluid resuscitation likely contributed to ARDS risks for large TBSA patients. Our data suggest that strategies to optimize fluid administration are important to improve patient outcomes, but should focus on clinical parameters rather than calculated fluid goals.

**Keywords:** major burns; fluid resuscitation; outcomes; retrospective analysis

## 1. Introduction

Prompt and adequate fluid resuscitation is considered essential for survival in patients presenting with greater than 20% TBSA (total body surface area) thermal burns [1]. Under-resuscitation has been associated with worsened organ failure and increased mortality [2,3]. To promote adequate fluid resuscitation in burn patients, clinicians have traditionally used a calculation based on a patient's weight and TBSA burn size to target a fluid resuscitation goal in the first 24 h. This original formula was developed more than 50 years ago, and calculates a targeted fluid rate at 4 mL/kg/%TBSA administered over 24 h (with the first half of that volume in the first 8 h) [4].

It has been increasingly recognized, however, that too much fluid resuscitation can be associated with complications in burn victims. These complications include the development of ARDS (acute respiratory distress syndrome), pneumonia, and compartment syndrome [5,6]. In 2008, the American Burn Association published practice guidelines that decreased the recommended fluid goal in the first 24 h to the 2–4 mL/kg/%TBSA range. This guideline acknowledged the insufficient evidence to make a strong recommendation regarding an initial fluid rate or type of fluid, but modified the calculation to help reduce the risks of over-resuscitation. The guideline additionally suggested that subsequent fluid management during the first 24 h should be adjusted to achieve a urine output of 0.5–1 cc/kg/h.

The objective of this study was to retrospectively assess the volume of fluids administered in the first 24 h to patients with greater than 20% TBSA burns over a 7.5-year period in a specialized Burn Intensive Care Unit (ICU). This included 4 years before and 3.5 years after a reduction in the targeted fluid goal (from 4 mL/kg/%TBSA to 2 mL/kg/TBSA) was incorporated into the standard resuscitation strategy in this ICU. The study evaluated the impact of this change on fluids received in the first 24 h before and after this change. Additionally, exploratory analyses evaluated the correlation of total fluids received and the type of fluid received with mortality or other secondary outcomes, paying particular attention to either over- or under-resuscitation.

## 2. Materials and Method

Medical charts of all patients admitted to the Grossman Burn Center with >20% TBSA burns from January 2014 to August 2021 were examined. The Grossman Burn Center is a 22-bed specialized burn unit located near Los Angeles, CA, USA. The data were compiled to record the age, weight, type of burns, time of injury, and percentage of total body surface area involved. The volume and type of fluid given at 24 and 48 h from the onset of injury were noted. ICU length of stay, hospital length of stay, development of pneumonia, acute renal failure, ARDS within one week of the onset of thermal injury, and mortality were also recorded.

### 2.1. Fluid Resuscitation Protocol

The fluid resuscitation protocol was nurse-driven, with physician input. From 2014–2018, the resuscitation goal was 4 mL/kg/%TBSA within the initial 24 h, which was then changed to 2 mL/kg/%TBSA at the end of 2018, except in cases of electrical burns. The protocol allowed for an increase and decrease in the lactated Ringer's rate, aiming for a urine output goal of 30–50 mL/h. Albumin could also be ordered, at the discretion of the attending physician, both for difficult resuscitation and to decrease the overall fluid volume, within the first 24 h. The albumin protocol started with 5% albumin at one-third of the hourly fluid resuscitation rate, preferably 10 h after the burn. Albumin boluses could also be utilized as rescue measures, at the physician's discretion.

### 2.2. Inhalational Injury

All of the burn patients with suspected inhalational injury (intubated prior to arrival to the burn center, rescued from fire in an enclosed space, with the presence of facial, perioral or nasal burns) underwent bronchoscopy within the first 48 h after sustaining burns. Inhalation injury was confirmed based on presence of carbonaceous deposits and/or erythema in the airways.

### 2.3. Statistical Analysis

The aforementioned data were compiled in a spreadsheet. For univariate analysis, a Student's *t*-test was performed on continuous variables that followed a Gaussian distribution, while the Wilcoxon test was applied if the distribution was not Gaussian. Fisher's exact test was utilized to test differences in proportions. For multivariate analysis, logistic regression was used if the outcome was discrete, without a time component. For survival

analysis, the Cox proportional hazards model was used. For continuous variables, linear regression was used to develop the multivariate model. In multivariate models, we used a feature for approximately every ten cases to prevent overfitting the model. In exploratory analyses, we evaluated the further additional features of interest. In particular, we aimed to explore whether under and over-resuscitation were associated with mortality and ARDS development. In addition, we wanted to investigate the role of albumin administration in these outcomes.

## 3. Results

A total of 104 patients with more than 20% TBSA burns were admitted to the Grossman Burn Center ICU from January 2014 to August of 2021. Some 4 patients were excluded, as they were transferred to our facility more than 48 h from the onset of burns. An additional 5 patients had no records of the type and volume of fluid received before transfer to our facility, leaving 95 patients to be included in this analysis.

### 3.1. Demographics

Table 1 shows the demographic data. Of the 95 patients included in this analysis, 76 patients (80%) survived admission. Some 13 of the 95 patients (13.7%) were admitted to the burn ICU from the emergency department, and the remainder were transferred from outside medical centers. The median age of the patients was 41 years (range: 17–90 years), and 73 of the 95 patients were men (76.8%). The median TBSA, calculated at our facility, was estimated at 36% (range: 19–97%). All patients (100%) analyzed received Ringer's lactate as primary fluid for resuscitation, and 40 of the 95 patients (42%) received albumin (colloids) as a rescue infusion within 24 h of injury. Some 44 of the 95 patients (46.3%) had a concurrent inhalational injury. The median length of hospital stay was 37 days (range: 1–165 days) and the median ICU length of stay was 18 days (range: 1–95 days). A total of 17 of the 95 patients developed ARDS (17.9%), 51 of the 95 (53.7%) patients developed pneumonia, and 34 of the 95 patients (35.8%) developed acute renal failure within the first 7 days of admission.

**Table 1.** Demographics.

| Feature | Data |
| --- | --- |
| Date Range | 28 February 2014–28 August 2021 |
| Total number of admissions | 95 |
| Number that survived admission | 76 (80.0%) |
| Age (years) (median: range) | 41 (range: 17–90) |
| Number men (%) | 73 (76.8%) |
| Weight (kg) (median) | 82 (range: 37–157) |
| TBSA (median: range) | 36 (range: 19–97) |
| Lactated Ringer's given (number of patients) | 95 (100%) |
| Normal saline given (number of patients) | 14 (14.7%) |
| Albumin given (number of patients) | 37 (38.9%) |
| Colloids given (number of patients) | 40 (42.1%) |
| Inhalational injury (number) (%) | 44 (46.3%) |
| Electrical burn (number) (%) | 4 (4.2%) |
| Number weekday admission (%) | 76 (80%) |
| Number ED admission (%) | 13 (13.7%) |
| Hospital length of stay (median) | 37 (range: 1–165) |

**Table 1.** *Cont.*

| Feature | Data |
|---|---|
| ICU length of stay (median) | 18 (range: 1–95) |
| ARDS (%) | 17 (17.9%) |
| Pneumonia first 7 days | 51 (53.7%) |
| AKI first 7 days | 34 (35.8%) |

*3.2. Analysis of the Amount of Fluid Received during Resuscitation*

Table 2 exhibits the median fluid resuscitation in mL/kg/%TBSA with minimum and maximum ranges for each year. From 2014 to 2018, 4 mL/kg/%TBSA burn formula was used for initial fluid resuscitation in the first 24 h. By the end of 2018, the ICU transitioned to a 2 mL/kg/%TBSA formula for fluid a 24 h fluid goal (unless there was an electrical burn, in which case the 4 cc/kg formula was utilized). However, despite this change, the median fluid administered from 2019 onwards remained close to 4 mL/kg/%TBSA. There was no statistical difference in terms of the amount of resuscitation that the patients received from year to year, despite the change in fluid resuscitation goals.

**Table 2.** Fluid distribution by year (not statistically significant).

| Fluids by Year (Not Statistically Different) | | | | |
|---|---|---|---|---|
| Year | Median (mL/kg/TBSA) | Minimum (mL/kg/TBSA) | Maximum (mL/kg/TBSA) | *p*-Value |
| 2014 | 3.66 | 1.82 | 6.84 | referent group |
| 2015 | 4.35 | 2.36 | 9.04 | 0.131 |
| 2016 | 4.12 | 2.92 | 10.93 | 0.232 |
| 2017 | 2.86 | 1.41 | 4.88 | 0.202 |
| 2018 | 3.72 | 2.75 | 6.72 | 0.796 |
| 2019 | 3.58 | 2.08 | 8.69 | 0.949 |
| 2020 | 3.84 | 1.04 | 6.43 | 0.611 |
| 2021 | 3.79 | 1.13 | 8.63 | 0.845 |

*3.3. Analysis of Fluid Resuscitation by Weight Quartiles*

The fluid resuscitation protocol, after the initial calculation based upon ABA recommendations, was based on clinical parameters such as a urine output of 30–50 cc/h and blood pressure. Comparing fluid administration within the first 24 h to the admission weight of our patients, lighter patients received significantly more fluid/kg than the heavier patients (Table 3 and Figure S1)

**Table 3.** Initial fluid resuscitation by weight quartiles.

| Weight Quartiles | Median (mL/kg/%TBSA) | 1st Quartile of Fluids (mL/kg/%TBSA) | 3rd Quartile of Fluids (mL/kg/%TBSA) |
|---|---|---|---|
| 1st quartile (range: 37.0 kg to 72.5 kg) | 4.66 | 3.47 | 4.96 |
| 2nd quartile (range: 72.5 kg to 82.8 kg) | 4.33 | 2.94 | 5.65 |
| 3rd quartile (range: 82.8 kg to 92.0 kg) | 3.91 | 2.63 | 4.83 |
| 4th quartile (range: 92.0 kg to 157.0 kg) | 3.21 | 2.10 | 3.89 |

### 3.4. Mortality

Table 4 represents the univariate analysis comparing different variables between patients who survived and those that died. Higher age and a greater involvement of TBSA were both associated with mortality. These were similarly demonstrated by the Kaplan–Meier survival curves (Figures 1 and 2). Subsequent multivariate analyses confirmed that both increased age and increased TBSA were associated with increased mortality, reflected in both the Cox proportional hazards analysis (Table S1) and logistic regression analysis (Table S2). Additionally, upon univariate analysis (Table 4), lower systolic blood pressure (SBP), diastolic blood pressure (DBP), temperature, SpO$_2$, and bicarbonate levels (HCO3) upon admission were associated with increased risk of mortality. Similarly, higher respiratory rate (RR), creatinine and elevated lactate upon admission were associated with death. A lower admission heart rate (HR) and hemoglobin level were also associated with increased mortality. Documented evidence of inhalational injury was linked to increased mortality.

**Table 4.** Mortality: univariate analysis.

| Features | Survived | Died | *p*-Value |
|---|---|---|---|
| Age (years) (median) | 40 | 58 | <0.0001 |
| TBSA (median) | 34 | 54 | <0.0001 |
| Percentage difference between first 24 h intake compared to ABA required fluids (median) | 84.13 | 82.36 | <0.0001 |
| Net I/O balance first 24 h in mL/kg/TBSA (median) | 3.74 | 3.65 | <0.0001 |
| SBP on admission (median) | 123 | 107 | <0.0001 |
| DBP on admission (median) | 66 | 59 | <0.0001 |
| HR on admission (mean) | 102 | 98 | <0.0001 |
| RR on admission (median) | 20 | 23 | <0.0001 |
| Spo2 on admission (median) | 99.5 | 98 | <0.0001 |
| Temperature (Celsius) on admission (median) | 37.22 | 37.17 | <0.0001 |
| Creatinine on admission (median) | 0.93 | 1.41 | <0.0001 |
| Hemoglobin on admission (mean) | 15.36 | 14.35 | <0.0001 |
| Lactate on admission (median) | 3.85 | 4.3 | <0.0001 |
| HCO3 on admission (mean) | 22.04 | 19.83 | <0.0001 |
| Proportion survived | Feature = 0 | Feature = 1 | *p*-Value |
| Gender (Male = 1) | 0.77 | 0.81 | 0.76 |
| Inhalational injury = 1 | 0.88 | 0.7 | 0.04 |
| Weekday admission = 1 | 0.95 | 0.76 | 0.11 |
| ED admission = 1 | 0.79 | 0.85 | 1 |

In multivariate analyses from Tables 5 and 6, death was associated with age and TBSA involvement. Further exploratory analyses (Figure 3) suggest that under-resuscitation may be associated with increased mortality, though this did not reach statistical significance. Including albumin administration in the multivariate analysis (Table 5) revealed that albumin administration was associated with increased mortality along with under-resuscitation, though this was not statistically significant. When inhalational injury was included as a predictor for death (Table 6), our calculations demonstrated that inhalation injury was associated with increased mortality along with albumin administration.

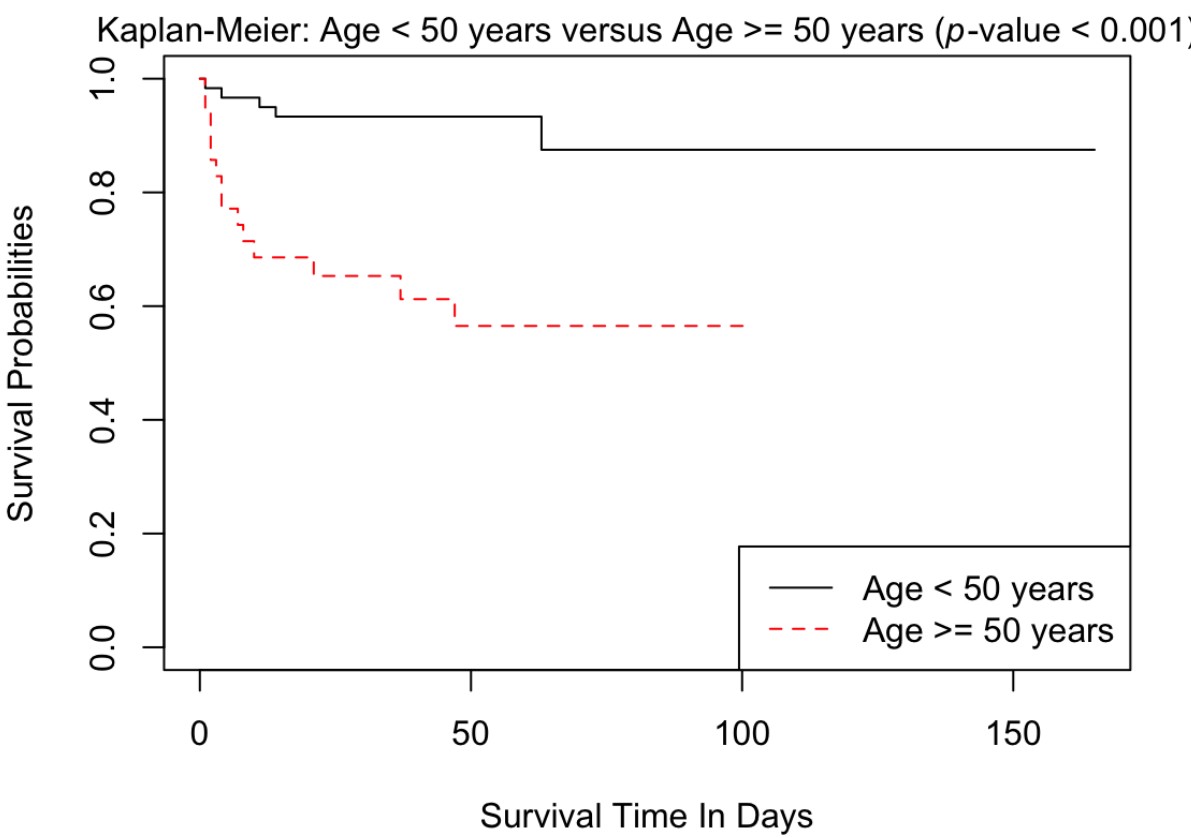

**Figure 1.** Kaplan–Meier curves comparing patients <50 years to those patients ≥50 years.

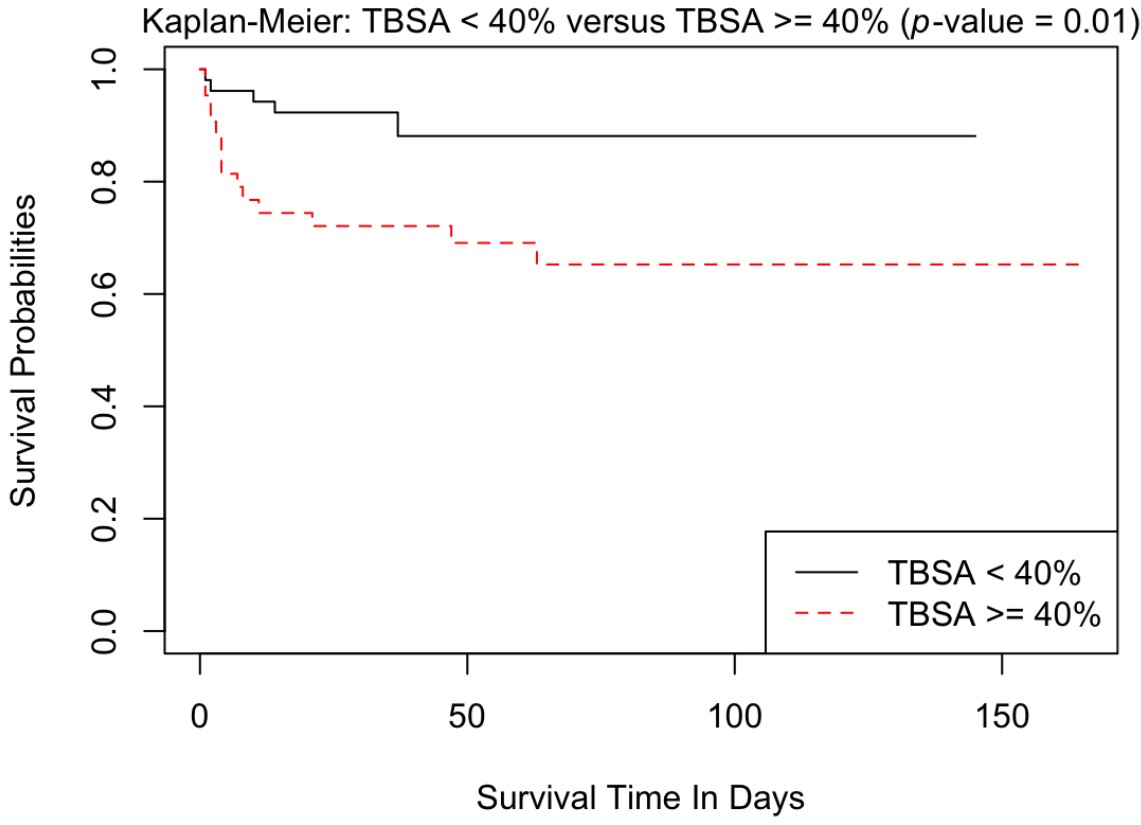

**Figure 2.** Kaplan–Meier curves comparing patients with <40% TBSA to those with ≥40% TBSA.

In terms of fluid resuscitation, those that survived had a higher initial fluid resuscitation in the first 24 h (Figure 3), though both those that survived and those that died had a median initial fluid resuscitation that was above 3 mL/kg/%TBSA.

**[Boxplot] Death: Fluids in 1st 24 hours (mL/kg/TBSA)**
Death = 1   *p*-value < 0.0001
Date Range:
2014-02-28 04:00:00
2021-08-28 22:40:00

**Figure 3.** Box plot of initial resuscitation comparing patients that died to those that survived.

**Table 5.** Exploratory survival analysis comparing under- versus over-resuscitation by 25% percent difference, compared to predicted fluid in first 24 h with albumin (in liters).

| Feature | Odds Ratio | *p*-Value |
|---|---|---|
| Age (in years) | 1.10 | <0.0001 |
| TBSA | 1.08 | <0.0001 |
| Over-resuscitation by more than 25% of ABA in first 24 h | 2.90 | 0.33 |
| Under-resuscitation by less than 25% of ABA in first 24 h | 17.95 | 0.05 |
| Albumin given (in liters) | 1.46 | 0.01 |

**Table 6.** Exploratory survival analysis comparing under- versus over-resuscitation by 25% percent difference, compared to predicted fluid in first 24 h with albumin (in liters) and inhalational injury.

| Feature | Odds Ratio | *p*-Value |
|---|---|---|
| Age (in years) | 1.10 | <0.0001 |
| TBSA | 1.08 | <0.0001 |
| Over-resuscitation by more than 25% of ABA in first 24 h | 1.79 | 0.61 |
| Under-resuscitation by less than 25% of ABA in first 24 h | 9.07 | 0.15 |
| Albumin given (in liters) | 1.65 | <0.05 |
| Inhalational injury | 3.47 | <0.05 |

*3.5. ARDS Development*

We defined acute respiratory distress syndrome (ARDS) using the following criteria:

Lung injury of acute onset within 1 week of admission post-burn;

Bilateral opacities on chest imaging;

Respiratory failure not explained by heart failure.

The following severity classification of decreased arterial $PaO_2/FiO_2$ ratio was used:

Mild ARDS: 201–300 mmHg;

Moderate ARDS: 101–200 mmHg;

Severe ARDS: $\leq$100 mmHg.

Inhalational injury was diagnosed using the circumstance of burn injury (closed space), clinical manifestations (facial burn, soot in mouth or pharynx, hoarseness or carbonaceous material), and confirmation by flexible bronchoscopy performed during the first 3 days of admission.

Table 7 represents the univariate analysis comparing those patients that developed ARDS to those who did not. As with mortality, ARDS development was associated with increased age and TBSA. Compared with those that did not develop ARDS, increased fluid intake within the first 24 h was associated with development of ARDS. When calculating fluid requirements based upon the modified Brooke fluid formula of 2 mL/kg/% TBSA, those individuals who developed ARDS had a higher percentage of being over the predicted resuscitation volume and on average, received twice the volume predicted in the first 24 h. Lower SBP, DBP and $HCO_3$ upon admission were associated with increased risk of ARDS. Higher RR, creatinine, and lactate levels upon admission were associated with increased risk of ARDS. A higher proportion of patients with inhalational injury developed ARDS, although the difference was not significant. Those patients who developed ARDS also had a higher median initial fluid resuscitation (Figure 4).

**Table 7.** ARDS development: univariate analysis.

| Features | No ARDS | ARDS | *p*-Value |
|---|---|---|---|
| Age (years) median | 41 | 53 | <0.0001 |
| TBSA (%) median | 34 | 48 | <0.0001 |
| Net I/O balance first 24 h in mL (median) | 8595 | 15734 | <0.0001 |
| Intake in first 24 h in mL (median) | 9783.5 | 16100 | <0.0001 |
| Percentage difference between first 24 h intake compared to ABA fluids requirement (median) | 82.20 | 106.06 | <0.0001 |
| SBP on admission (median) | 123 | 117 | <0.0001 |
| DBP on admission (median) | 67 | 61 | <0.0001 |
| HR on admission (mean) | 102 | 97 | <0.0001 |
| RR on admission (median) | 20 | 24 | <0.0001 |
| Spo2 on admission (median) | 99 | 99 | <0.0001 |
| Temperature (Celsius) on admission (median) | 37.2 | 37.3 | <0.0001 |
| Creatinine on admission (median) | 1.02 | 1.33 | <0.0001 |
| Hemoglobin on admission (mean) | 15.1 | 15.3 | <0.0001 |
| Lactate on admission (median) | 3.55 | 4.3 | <0.0001 |
| HCO3 on admission (mean) | 21.89 | 20.23 | <0.0001 |
| Proportion that developed ARDS | Feature = 0 | Feature = 1 | *p*-Value |
| Gender (male = 1) | 0.27 | 0.15 | 0.23 |
| Inhalational injury = 1 | 0.14 | 0.23 | 0.29 |
| Weekday admission = 1 | 0.16 | 0.18 | 1 |
| ED admission = 1 | 0.2 | 0.08 | 0.45 |

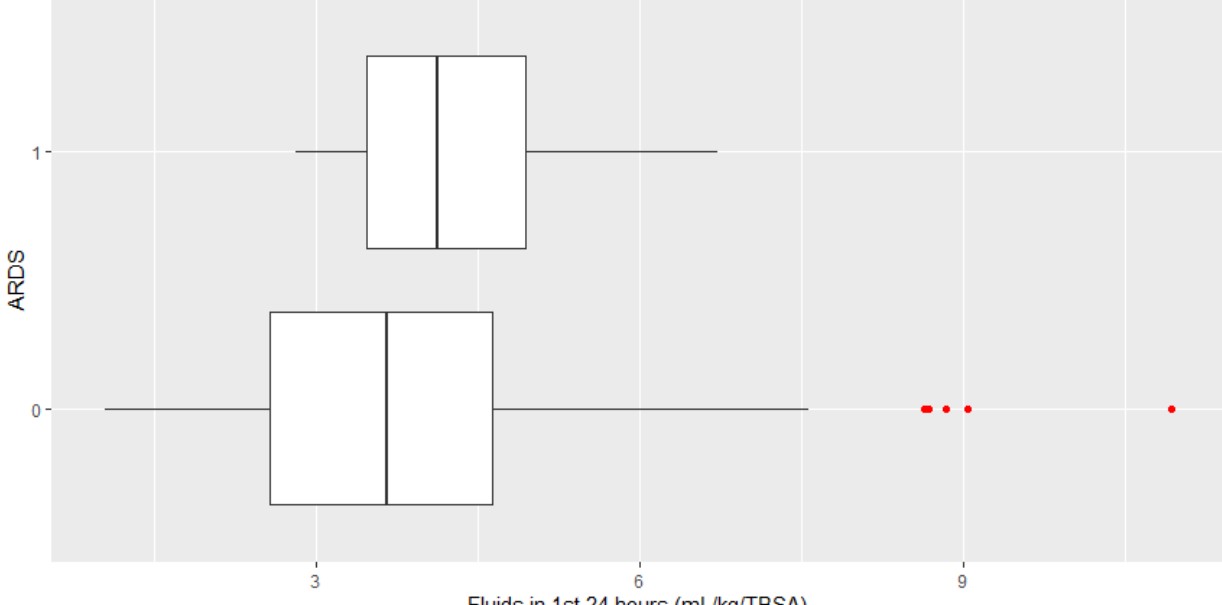

**Figure 4.** Box plot of initial fluid resuscitation comparing those patients that developed ARDS to those that did not.

Unlike for survival analysis, we did not find a statistically significant association between albumin administration and development of ARDS upon multivariate analysis (Table 8).

**Table 8.** Exploratory analysis for ARDS development including first 24 h intake as mL/kg/%TBSA, and albumin.

| Feature | Odds Ratio | *p*-Value |
|---|---|---|
| Age (in years) | 1.03 | <0.05 |
| TBSA | 1.04 | <0.05 |
| First 24 h intake as mL/kg/TBSA | 1.04 | 0.81 |
| Albumin given | 2.39 | 0.14 |

### 3.6. Albumin Administration and Its Association with Initial Fluid Administration

Upon univariate analysis, albumin administration was associated with increased initial fluid resuscitation (Figure 5). Here, the fluids referenced in the initial resuscitation period looked at only the non-albumin fluids. This was because there were some patients who received in the order of 1000–5000 mL of albumin as part of the initial resuscitation.

Factoring this in, there was a suggestion that albumin given in the first 24 h was associated with worse survival, as noted above in Tables 5 and 6. While not shown here, albumin administration did not improve urine output upon multivariate analysis.

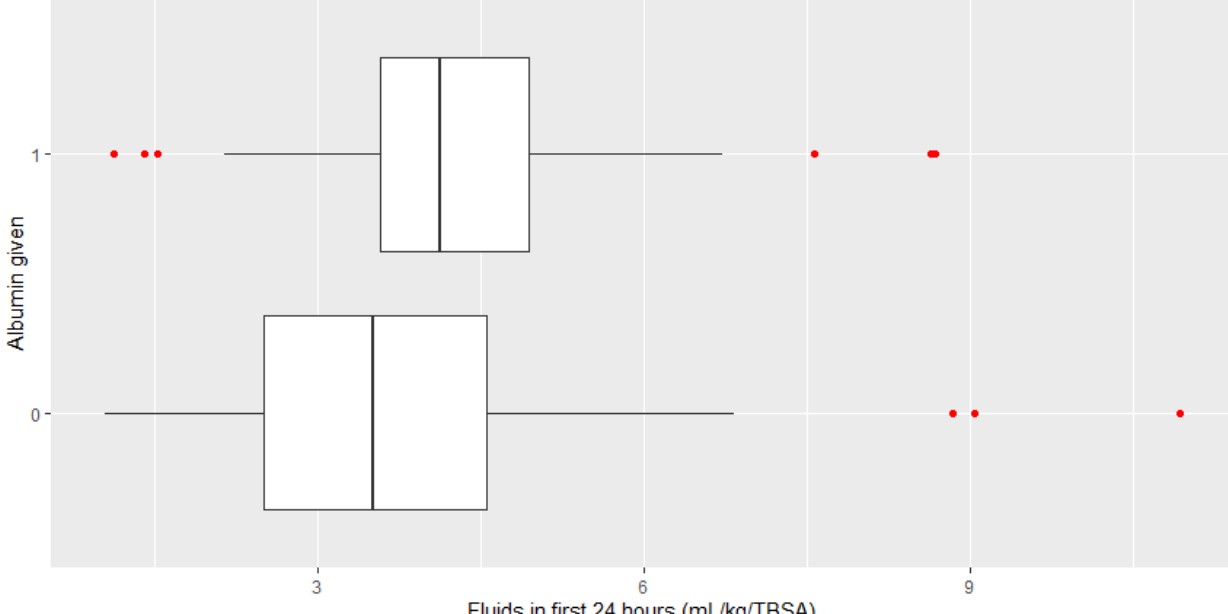

**Figure 5.** Box plot of initial fluid resuscitation comparing those patients that received albumin compared to those that did not.

### 3.7. Development of Pneumonia

The diagnosis of pneumonia was made using standard Centers for Disease Control criteria, including new or changing infiltrate on chest radiograph, leukocytosis, temperature > 38.5 °C (100.4 °F) or hypothermia < 35.0 °C (95 °F), and the presence of purulent tracheal aspirates. In order to prevent ventilator-associated pneumonia, the following measures are taken in our Burn Care Unit: elevating the head of the bed, oral decontamination, subglottal suctioning, and early intervention with chest physiotherapy. As noted in Table 1, 51 of the 95 patients (53.7%) developed pneumonia within the first week after the injury. Table 9 demonstrates that while age, percentage surface area burns, and volume of fluid administrated were not associated with pneumonia, inhalational injury increased the risk of developing pneumonia by more than 2-fold.

**Table 9.** Logistic regression analysis of the development of pneumonia in the first 7 days.

| Feature | Odds Ratio | *p*-Value |
|---|---|---|
| Age (in years) | 0.99 | 0.50 |
| TBSA | 0.99 | 0.32 |
| First 24 h intake as mL/kg/TBSA | 1.07 | 0.61 |
| Inhalational injury | 2.68 | 0.04 |

### 3.8. ICU Length of Stay

Upon multivariate analysis (Table 10), TBSA was associated with increased ICU length of stay. Factoring in sedation into this analysis (Table 11) did not change results and was not statistically significant.

**Table 10.** ICU length of stay.

| Feature | Coefficient | *p*-Value |
| --- | --- | --- |
| Age (in years) | −0.01 | 0.93 |
| TBSA | 0.21 | <0.05 |

**Table 11.** Exploratory analysis: ICU length of stay with sedation.

| Feature | Coefficient | *p*-Value |
| --- | --- | --- |
| Age (in years) | 0.06 | 0.68 |
| TBSA | 0.22 | <0.05 |
| Fentanyl rate 24 h | 0.00056 | 0.71 |
| Propofol rate 24 h | 0.0053 | 0.28 |
| Midazolam rate 24 h | 0.01 | 0.87 |

*3.9. Comparing Mortality and Outcomes before and after the Change in Fluid Resuscitation Goals*

We compared the mortality and rate of ARDS and AKI before and after the change in fluid resuscitation goals and found no statistical difference in the outcomes (Table S3).

## 4. Discussion

This study evaluated consecutive admissions of large TBSA burns to an adult burn ICU over a 7.5-year period. Analysis of our demographic data showed a majority of these patients were males. This is consistent with findings from other adult burn ICUs in the United States [7–9]. We also noted the majority of the admissions to our burn center were transferred from outside medical facilities, which is not surprising given that our center specializes in the comprehensive treatment of burns.

With regard to the etiology of burns, flame burns were the major cause of severe burns among our patients, with 46% of the patients having a concurrent inhalational injury. The incidence of inhalational injury in this study (46%) is higher than the range of 6–20% typically seen in burn ICUs [10]. The high rate of inhalational injury observed in this study may be related to the burn's mechanism, such as fire, which was a frequent cause in this sample. Additionally, patients in this study were older, with the median age being 41 years. Increased age and large TBSA burn size are known risks for developing inhalational injury [11,12]. The median length of hospital stay for our patients was 37 days over this time period.

When analyzing the fluids administered to the patients in our ICU during this period, we were surprised that despite changes in our initial fluid resuscitation formula, the median fluid administered to our patients did not change, and remained near 4 mL/kg/%TBSA over the first 24 h. Post implementation of 2 mL/kg X TBSA fluid resuscitation formula, the patients initially received less fluid but because the protocol allowed the nurses to increase the fluid rate for goal UOP of 30–50 cc/h, the patients based on our study ended up receiving the same total amount. Interestingly, the analysis revealed that lighter patients received significantly more fluids than the heavier patients per kilogram, suggesting urine output goals were not adjusted for weight. This is consistent with the resuscitation protocol using an absolute urine output as a clinical goal rather than a urine output per weight-based goal. In addition, there may be barriers, such as nursing staff being hesitant to decrease the resuscitation fluid rate even if the protocol indicates to do so, resulting in overall increased fluid resuscitation. Furthermore, given the high percentage of patients with inhalational injury in this study, the patients may have received more fluid than the level predicted by the initial calculation (which does not adjust for inhalational injury). We also found that the patients receiving albumin received more fluid. We postulate that this may have been because of ongoing clinical deterioration despite adequate fluid resuscitation, with

the use of albumin serving as salvage therapy. As such, it may not be that albumin usage is related to increased mortality, but that it serves as a surrogate for decompensation despite adequate fluid resuscitation. We compared the mortality and rate of ARDS and AKI before and after the change in fluid resuscitation goals and found no statistical difference in the outcomes, likely because the group pre and post change in resuscitation goals ended up receiving the same amount of fluid within the first 24 h of their burn injury

A univariate analysis of our data did show that increased mortality was associated with less fluid resuscitation (although this was still significantly higher than 2 mL/kg/%TBSA). While close to but not statistically significant, multivariate analysis noted an association between under-resuscitation and mortality, but did not find such an association between over-resuscitation and mortality.

Not surprisingly, increased age, high TBSA % burn, and inhalational injury were all associated with decreased survival in our data. This has been well documented in other studies of adult burn victims [13]. Patients with more severe burns had increased initial creatinine levels. While our numbers were not sufficient, upon multivariate analysis, acute kidney injury was independently associated with increased mortality. Upon exploratory analysis, we found that albumin administration within the first 24 h was associated with decreased survival; however, the time of albumin administration in the resuscitation course was not analyzed. Given our limited cases, we would consider this result a potential hypothesis-generating event for future studies, rather than a causal inference.

The study looked at both ARDS and pneumonia as possible secondary outcomes from excessive fluid administration. The classification of ARDS used the Berlin definition [14]. We found that increased TBSA burn and age were associated with the development of ARDS. This supports the role of tissue injury, inflammatory response, and burn shock in the possible genesis of ARDS [15–17]. Contrary to other studies [18,19], the presence of inhalational injury did not appear to be associated with development of ARDS in our patients. We did, however, find a relationship between fluid resuscitation volume and development of ARDS. Patients who received more than twice their predicted volume (2 mL/kg/BSA) within the first 48 h of burn injury were at higher risk of developing ARDS. Other studies have supported the benefit of limiting fluid administration to improve outcomes in ARDS patients in mixed populations of ICU patients [20,21]. Excessive fluids are thought to be harmful due to the increased permeability of the pulmonary membranes in this condition.

We did not find that pneumonia was correlated with excessive fluid volume administered early in the hospital course, but we did find that the presence of inhalational injury increased the risk of pneumonia by more than 2-fold within the first week of injury. Several other studies have found an association between pneumonia and the presence of inhalational injuries [22–25]. We did not find a statistical correlation between TBSA and pneumonia, however, as has been seen in some of these studies [23]. It is possible that the high incidence of pneumonia in our sample (53.7%) could be attributed to an over diagnosis, as the criteria defining pneumonia are not specific enough to differentiate it from tracheobronchitis, or the radiographic changes typically seen with smoke inhalational injury alone.

The main limitation of the present study is that it was a single-center study with a relatively small number of patients. Using absolute urine output as a resuscitation goal rather than weight-based urine output is another major limitation of the study. The small sample size might have underestimated the effect of risk factors relevant for the studied outcomes. Furthermore, we did not control for provider preferences in the study. There exists a core of both physicians and providers, and since they often cross-over for each other, we postulate that the care averages out statistically.

## 5. Conclusions

Initial intravenous fluid administration to a large thermal burn patient is a balance between over- and under-resuscitation. Our retrospective review of early resuscitation

of these patients in our ICU over 7 years did not show that changes in initial calculation of fluid requirements (using a traditional or revised formula) changed the actual fluids administered over 24 h. Fluid volume appeared to be guided by clinical criteria rather than initial calculation of anticipated fluid requirements. Our data did suggest that under-resuscitation was associated with increased mortality, and excessive IV fluids resulted in greater risk of ARDS. We could not demonstrate that albumin administration improved outcomes. Our study also suggested that inhalational injury predicted pneumonia, rather than fluid volume or size of thermal burn. Further attempts to optimize fluid administration during the early resuscitation of large TBSA burn patients should most likely focus on the bedside parameters (such as weight-based urine output) and possibly on the timing of colloid over crystalloid administration. Avoiding under-resuscitation remains critical, but excessive fluid resuscitation likely contributes to ARDS risks for large TBSA burn patients.

**Supplementary Materials:** The following supporting information can be downloaded at: https://www.mdpi.com/article/10.3390/ebj4020021/s1, Figure S1. Fluid distribution by weight; Table S1. Mortality Analysis Logistic Regression; Table S2. Survival Analysis Cox Proportional Hazards; Table S3. ARDS Development Logistic Regression.

**Author Contributions:** Conceptualization, M.M. and K.M.R.; Data curation, M.I.S. and S.C.; Formal analysis, M.M., S.C. and S.B.H.; Investigation, M.M., M.I.S., S.C. and K.M.R.; Methodology, M.M.; Supervision, P.H.G.; Validation, S.B.H.; Writing—original draft, M.M.; Writing—review and editing, M.I.S., S.C., S.B.H. and P.H.G. All authors discussed the results and contributed to the final manuscript. All authors have read and agreed to the published version of the manuscript.

**Funding:** This research received no external funding.

**Institutional Review Board Statement:** This study was approved by the UCLA institutional review board. The study was conducted in accordance with the Declaration of Helsinki, and approved by the Institutional Review Board University of California, Los Angeles (protocol ID IRB#21-001332 and date of approval was 25 August 2021).

**Informed Consent Statement:** Informed consent was waived as it was not feasible to contact these patients to obtain consent, as a few of these patients did not survive their hospitalization. Additionally, the patient demographics admitted to our institution included homeless people who would be very difficult to contact. The design of the study did not allow the possibility of obtaining consent, and the data collected was anonymized to protect the patients.

**Data Availability Statement:** Data will be available on request as they are saved on our institution's BOX platform and require a specific username and password to access them.

**Acknowledgments:** The authors thank all the doctors at the Grossman Burn Center and acknowledge their dedication in providing the best care to the patients at the Burn Center. We also want to thank Carolyn Greco for assisting us with collecting the data for this research.

**Conflicts of Interest:** The authors declare no conflict of interest.

## Abbreviations

TBSA—total body surface area, ARDS—acute respiratory distress syndrome, AKI—acute kidney injury, I/O—fluids in and out, SBP—systolic blood pressure, DBP—diastolic blood pressure, HR—heart rate, RR—respiratory rate, HCO3—bicarbonate, ED—emergency department, ABA—American Burn Association, ICU—Intensive Care Unit

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
