# Peer review of "Evaluation of Outcomes following Reduction in Targeted Fluid Administration in Major Burns"

_2673-1991, doi:10.3390/ebj4020021_

Round 1

Reviewer 1 Report

This is a well written single center retrospective study looking at outcomes following reduction of targeted fluid administration in major burns with target of 2 ml/kg/% burn instead of 4 ml/kg/% burn. 

There are some questions. 

40 of the 95 patients received colloids as rescue infusion.  Amount varies widely.  Was colloid used due to difficult resuscitation or to decrease volume? 

Physician driven or nurse driven resuscitation?  Was computer supported program used for resuscitation?

The study evaluated the impact of the change in fluids received in the first 24 hours before and after the change of reduction in targeted fluid goal.  From 2014 to 2018, 4 ml formula was used for initial fluid resuscitation in the first 24 hours.  By the end of 2018, they transitioned to a 2 mL formula for fluid a 24 hour fluid goal.  Instead of comparing 2 ml vs 4 ml, the authors bundle up all patients together in terms of looking at the outcome.  Can you compare the two groups pre implementation of 2 ml vs post implementation of 2 ml and see if outcomes including mortality and secondary outcomes are different?

Did post implementation receive less fluid initially and then received more to catch up and at the end, received same total amount? 

How many had confirmed inhalation injury with bronchoscopy? Face burn and singed hair do not always mean you have inhalation injury. 

Why were some of them under-resuscitated if fluid volume was titrated to clinical criteria?

Is inhalation injury rate same for pre 2018 vs post 2018?

The authors stated that there might be barriers to decreasing fluid resuscitation rates once the urine output goal has been achieved.  There should be a protocol to decrease IV fluid when urine output is more than adequate.

Was urine output target close to 0.5ml/kg/hr or close to 1ml/kg/hr?  Most aim for 0.5 to decrease over resuscitation.

Do you have protocol for resuscitation using albumin or is it more individual patient based?

Is there a guideline to decrease IV fluid if urine output adequate?

Reviewer 2 Report

This is a valuable study that aims to address significant clinical questions related to fluid administration strategies and patient outcomes. By investigating various factors and their correlations, the study contributes to a deeper understanding of optimal treatment approaches in clinical practice

The study investigates two distinct fluid administration strategies under clinical guidance.

  1. In my opinion, there are additional questions that should be addressed to further enhance our understanding.

  2. For albumin resuscitation, was it continuous administration or "albumin rescue" initiated when urine output dropped and couldn't be improved by increasing fluids according to the provided guidelines?

  3. Were the outcomes different when compared to the use of colloids?

  4. Did fluid requirements vary between patients receiving preclinical albumin and those administered albumin as a "rescue procedure" when urine output stalled?

  5. Did patients receive either albumin or colloids exclusively, or were there cases where both were administered?

  6. Was there a correlation between third-degree burns and initial creatinine levels?

  7. How many patients required mechanical ventilation? Since mechanical ventilation can increase fluid demand, did outcomes correlate with its use? (Reference: Mackie, David P., Spoelder, Ed J., Paauw, Roel J., Knape, Paul, Boer, Christa: Mechanical ventilation and fluid retention in burn patients.)

Reviewer 3 Report

Thank you for allowing me to review your paper.

1. Did you control for provider preferences or do the providers in your ICU follow the protocol strictly?

2. Did you speak about whether the change to calculate your initial rate at 2 instead of 4 and whether this affected rates of acute kidney injury in these patients? 

3. Were ventilator days affected by coefficient of resuscitation? Were ventilator days in general affected in the pre- and post-alteration of resuscitation strategy groups?

Round 2

Reviewer 2 Report

Thank you, congratulations! 

Author Response

Thank you for your insightful feedback on our manuscript.

Reviewer 3 Report

Edits are satisfactory

Author Response

(The authors gave the same response as above.)
